

# PBL height estimation based on lidar depolarisation measurements (POLARIS)

Juan Antonio Bravo-Aranda[1,2,a], Gregori de Arruda Moreira[3], Francisco Navas-Guzmán[4], María José
Granados-Muñoz[1,2,b], Juan Luís Guerrero-Rascado[1,2], David Pozo-Vázquez[5], Clara Arbizu-Barrena[6],
Francisco José Olmo Reyes[1,2], Marc Mallet[7,c] and Lucas Alados Arboledas[1,2]

[1]Andalusian Institute for Earth System Research (IISTA-CEAMA), Granada, Spain
[2]Dpt. Applied Physics, University of Granada, Granada, Spain
[3]Institute of Energetic and Nuclear Research (IPEN), São Paulo, Brazil
[4]Institute of Applied Physics (IAP), University of Bern, Bern, Switzerland
[5]Dpt. of Physics, University of Jaén, Jaén, SCpain
[6]Laboratoire d'Aérologie, Toulouse, France
[7]Centre National de Recherches Météorologiques, Toulouse, France
[a]now at: Institute Pierre-Simon Laplace, CNRS-Ecole Polytechnique, Paris, France
[b]currently at: Table Mountain Facility, NASA/Jet Propulsion Laboratory, California Institute of Technology, Wrightwood,
California, USA
[c]now at: CNRM, Météo-France-CNRS, Toulouse, France

*Correspondence to*: Juan A. Bravo-Aranda (jabravo@ugr.es)

**Abstract.**

The automatic and non-supervised detection of the planetary boundary layer height ($z_{PBL}$) by means of lidar measurements
was widely investigated during the last years. Despite the considerable advances achieved the experimental detection still
present difficulties either because the PBL is stratified (typically, during night-time) either because advected aerosol layers are
coupled to the PBL. The coupling uses to produce an overestimation of the $z_{PBL}$. To improve the detection even in these
complex atmospheric situations, we present a new algorithm, called POLARIS (PBL height estimatiOn based on Lidar
depolARISation). POLARIS applies the wavelet covariance transform (WCT) to the range corrected signal and to the
perpendicular-to-parallel signal ratio (δ) profiles. Different candidates for $z_{PBL}$ are chosen and the attribution is done, based on
the WCT applied to the RCS and the δ. We use two ChArMEx campaigns with lidar and microwave radiometer (MWR),
conducted on 2012 and 2013, for the POLARIS' adjustment and validation. POLARIS improves the $z_{PBL}$ detection thanks to
the consideration of the relative changes in the depolarization capabilities of the aerosol particles in the lower part of the
atmospheric column. Taking the advantage of a proper determination of the $z_{PBL}$ determined by POLARIS and by MWR under
Saharan dust events, we compare the POLARIS and MWR $z_{PBL}$ with the $z_{PBL}$ provided by the Weather Research and
Forecasting (WRF) numerical weather prediction model. WRF underestimates the $z_{PBL}$ during daytime but agrees with the
MWR during night-time. The $z_{PBL}$ provided by WRF showed a better temporal evolution during daytime than during night-
time.





## 1 Introduction

The planetary boundary layer (PBL) is the region of the troposphere that is directly influenced by the processes at the Earth's surface, this region typically responds to surface forcing mechanisms with a time scale of about one hour or less (Stull, 1988). The PBL height, $z_{PBL}$, is a relevant meteorological variable with a strong effect on air pollution as it defines the atmospheric

volume that can be used for pollutant dispersion. Along the time, different approaches based on the use of elastic lidar data have been proposed for detecting the $z_{PBL}$ (e.g., Morille et al. 2007; Granados-Muñoz et al. 2012; Wang et al. 2012; Pal et al., 2013; Coen et al. 2014; Banks et al. 2015). Among them, some methods like the wavelet covariance transform (WCT) has already demonstrated to be a good tool for an automatic and unsupervised detection of the $z_{PBL}$ (Morille et al., 2007; Baars et al. 2008; Pal et al. 2010; Granados-Muñoz et al., 2012; Wang et al., 2012). This method can be considered the combination of

applying the so call gradient method to a Range Corrected profile after smoothing by a low-pass filter (Comerón et al. 2013). In these methods the top of the PBL is associated to the height where there is a sharp decrease of the RCS and thus of the aerosol load.

However, the experimental detection of the $z_{PBL}$ present challenges.. The PBL changes dynamically during the day and can

present different structures.. The diurnal period is characterized by a mixing layer (statically unstable), the so call convective boundary layer, Turbulent mixing controls the vertical dispersion up to the top of the CBL (Seibert 2000). The CBL is denominated mixed layer, when the homogenization is complete (neutral stability) something that happens when turbulence is really vigorous and there is an intense convection.. During night-time, we have the stable boundary layer (also known as nocturnal boundary layer) that is in direct contact with the surface, and the residual layer that is a region loaded with the aerosol

that reaches high elevation in the previous day's within the mixing layer (or mixed layer if was formed) (Stull, 1988). At night it is usual to have dry deposition of the aerosol particles due to the suppression of the convection that allows gravitational sedimentation and diffusion motion, among others (Seinfeld and Pandis, 1998), although some sporadic turbulence may exists (Stull et al., 1998). Sunrise and Sunset are characterized by the complexity of the PBL. Early in the morning sun irradiance contributes to the development of a mixing mixing layer (statically unstable) that co-exists with a stable boundary layer topped

by a residual layer (nearly statically neutral). The use of aerosol for the identification of the PBL height represents a challenge due to the PBL evolution and complex internal structure. Furthermore, coupling of aerosol layers that has been transported in the Free Troposphere with aerosol in the PBL or even the presence of clouds leads to under- or overestimation of the PBL height (Granados-Muñoz et al., 2012; Summa et al. 2013). Finally, the PBL can be strongly affected by the complexity of the underlying surface, this is the case for frozen surfaces or sea-inland interface (Stull et al., 1998). In this work, we present a

new method, called POLARIS (PBL height estimatiOn based on Lidar depolARISation), which is an ameliorated version of the method presented by Baars et al., (2008) and Granados-Muñoz et al. (2012). POLARIS uses the combination of the WCT applied to the RCS and the perpendicular-to-parallel signal ratio (δ) profiles. Using these profiles, we choose different candidates for the $z_{PBL}$ are chosen and performing the attribution through the POLARIS algorithm. POLARIS is particularly



useful when advected aerosol layers in the free troposphere are coupled to the PBL because the lidar depolarization ratio profiles provide information about the particle shape allowing the discrimination among different aerosol types. Furthermore, POLARIS improves the $z_{PBL}$ detection since the computation of δ (based on the ratio of two lidar signals) cancelled out the incomplete overlap effect allowing the $z_{PBL}$ detection at lower heights than using methods based exclusively on RCS (affected

by incomplete overlap). To simplify the nomenclature, hereafter, we will refer to the $z_{PBL}$ understanding the top of the mixing, mixed or residual layer except when needed.

The implementation and validation of POLARIS method use a data set of lidar and Micro Wave Radiometer measurements registered in ChArMEx (Chemistry-Aerosol Mediterranean Experiment, www.charmex.lsce.ipsl.fr) experimental campaigns during the summers of 2012 and 2013. ChArMEx is a collaborative research program federating international activities to

investigate Mediterranean regional chemistry-climate interactions (Mallet et al., 2016). One of the goals of ChArMEx is to reach a better knowledge on the atmospheric aerosol over the Mediterranean Basin (Dulac et al., 2014; Sicard et al., 2016; Granados-Muñoz et al., 2016).This works also contributes to the Mediterranean studies (as those carried out in the framework of  ChArMEx) since POLARIS allows the PBL detection under the frequent dust outbreaks affecting this region.

Since the experimental detection of $z_{PBL}$ is spatially and temporally limited due to instrumental coverage, the use of Numerical

Weather Prediction (NWP) models for the estimation of $z_{PBL}$ is feasiable alternative. At this regards, several validation studies of these model estimations have been conducted based on lidar and surface and upper air measurements (Dandou et al., 2009; Helmis et al, 2012), some of them in areas close to the study region (Borge et al., 2008; Banks et al., 2015). Results showed that NWP estimations of the $z_{PBL}$ are feasible and reliable, but with a tendency to the underestimation of the $z_{PBL}$ in most synoptic conditions. These model estimations of $z_{PBL}$ are also a key parameter for aerosol dispersion models. In this work the

WRF (Weather Research and Forecasting) NWP model (Skamarock et al., 2008), $z_{PBL}$ estimations are tested based on the POLARIS algorithm. Some of the period here tested include stringent conditions, as the presence of an advected aerosol layer coupled to the PBL.

## 2 Experimental site and instrumentation

In this work we use measurements registered in the Andalusian Institute for Earth System Research (IISTA-CEAMA). This

center is located at Granada, in Southeastern Spain (Granada, 37.16°N, 3.61°W, 680 m asl).The metropolitan Granada's population is around 350 000 inhabitants: 240 000 inhabitants from the city and 110 000 inhabitants from the main villages surround the city(www.ine.es). It is a non-industrialized city surrounded by mountains (altitudes up to 3479 m asl, Mulhacén peak).. Granada's meteorological conditions are characterized by large seasonal temperature differences (cool winters and hot summers) and by a rainy period between late autumn and early spring being the rain scarce the rest of the year.

The main local sources of aerosol particles are the road traffic, the soil re-suspension (during warm-dry season) and the domestic heating based on fuel oil combustion (during winter) (Titos et al. 2012). Additionally, due to its proximity to the African continent, Granada's region is frequently affected by outbreaks of Saharan air masses becoming an exceptional place





to characterize the Saharan dust. Additionally, Lyamani et al. 2010; Valenzuela et al., 2012 point to the Mediterranean basin as an additional source of aerosol particles in the region.

MULHACEN is a multiwavelength lidar system with a pulsed Nd:YAG laser, frequency doubled and tripled by Potassium Dideuterium Phosphate crystals. MULHACEN emits at 355, 532 and 1064 nm (output energies per pulse of 60, 65 and 110 mJ, respectively) and registers elastic channels at 355, 532 and 1064 nm and Raman-shifted channels at 387 (from N2), 408 (from H2O) and 607 (from N2) nm. The laser beam also passes through two beam expanders reducing the divergence and increasing the surface of the laser beam by a factor ×5 and ×4.5 for 355 nm and 532/1064 nm, respectively. The full overlap is reached around 1220 m agl although the overlap is complete at 90% between 520 and 820 m agl for all the wavelengths (Navas-Guzmán et al ., 2011; Rogelj, 2014). Further details are provided by Guerrero-Rascado et al. (2008, 2009).

In addition, a ground-based passive microwave radiometer (RPG-HATPRO, Radiometer Physics GmbH) continuously measured tropospheric temperature and humidity profiles during the studied period. The passive MWR performs zenith measurements of the sky brightness temperature with a radiometric resolution between 0.3 and 0.4 K root mean square error at 1-s integration time (Navas-Guzmán, 2014). The radiometer uses direct detection receivers within two bands: 22-31 GHz (providing information about the tropospheric water vapour profile) and 51-58 GHz (related to the temperature profile). In addition, surface meteorological data are also available from a co-located meteorological station. Temperature profiles are retrieved from surface meteorological and the brightness temperature measured at the V-band frequencies, where the first 3 frequencies are only used in zenith pointing (51.26, 52.28 and 53.86 GHz) and the last 4 (54.94, 56.66, 57.3 and 58 GHz) are considered for all the elevation angles (Meunier et al., 2013). The inversion algorithm is based on neural networks (Rose et al., 2005) trained using the radiosonde database of the Murcia WMO station nr. 08430 located at 250 km from Granada. The accuracy of the temperature profiles is0.8 K within the first 2 km and 1.5 K between 2 and 4 km. Vertical resolution increases with height: 30 m on the ground, 50 m between 300-1200 m, 200 m between 1200 and 5000 m and 400 m above. Navas-Guzmán, 2014).The MWR temperature profile is used to locate the $z_{PBL}(z_{PBL}^{MWR})$by two algorithms according to the characteristics of potential temperature profile: under convective conditions, fuelled by solar irradiance absorption at the surface and the associated heating, the parcel method is applied (Holzworth, 1964). Granados-Muñoz et al., 2012 already validated this methodology with radiosonde measurements. Conversely, under stable situations, $z_{PBL}^{MWR}$is obtained from the first point where the gradient of potential temperature (θ) is equal zero. Collaud-Coen et al. (2014) give further details about both methods. The uncertainty of the $z_{PBL}^{MWR}$is estimated to be 200 m below 2 km, and 400 m above 2 km because of the vertical resolution of the MWR temperature profile is between 100 and 500 m for heights below 3 km agl, where the PBL is usually located over Granada (Granados-Muñoz et al., 2012).



## 3 The POLARIS method

### 3.1 Wavelet Covariance Transform

The WCT, $W_F(a, b)$, applied to a generic function of height, $F(z)$, (e.g., RCS or δ) is defined as follows:

$$W_F(a,b) = \frac{1}{a} \int_{z_b}^{z_t} F(z) h\left(\frac{(z-b)}{a}\right) dz \qquad \text{Eq. 1}$$

where z is the height, $z_b$ and $z_t$ are the integral limits and $h((z - b)/a)$ is the Haar's function defined by the dilation, $a$, and the translation, $b$ (Fig. 1).

Fig. 2 shows an example of the WCT applied to the RCS ($W_{RCS}$). $W_{RCS}$ presents a maximum in coincidence with the sharpest decrease of the RCS and thus, the $W_{RCS}$ maximum is associated to a sharp decrease of the aerosol load which could be related to the top of the PBL. In this sense, Baars et al. (2008) proposed the use of the first maximum in the $W_{RCS}$ profile from surface

larger than a threshold value to detect the $z_{PBL}$. Granados-Muñoz et al. (2012) improved this method using an iterative procedure over the dilation parameter starting at 0.05 and decreasing with steps of 0.005. Baars et al. (2008) and Granados-Muñoz et al. (2012) provides a deeper analysis related to the wavelet method. However, this attribution cannot be generalized. The automatic application of these methods provides inappropriate attributions under complex scenarios in which aerosol load is stratified within the PBL or aerosol layers in the Free Troposphere are coupled to the PBL.

### 3.2 Description of POLARIS

POLARIS is based on the detection of the sharp decrease of the aerosol load with height by means of the range corrected signal together with the relative changes in the aerosol particle shape with height by means of the perpendicular-to-parallel signal ratio (δ): low δ values might related to spherical particle shape and vice versa (Gross et al., 2011). In this way POLARIS is able to detect the PBL height even when advected aerosol layers in the free troposphere are coupled to the PBL. Since

POLARIS is based on vertical relative changes, the depolarization calibration is not required and, thus, POLARIS can be applied to data from lidars not fully characterized. This increases the applicability of the method and facilitates the calculus.

POLARIS uses 10-min averaged range corrected signal (RCS) and perpendicular-to-parallel signal ratio (δ) and carries out the following steps:

1)    The WCT is applied to the RCS ($W_{RCS}$) and to δ ($W_\delta$). Then, both $W_{RCS}$ and $W_\delta$ signals are normalized respectively

to the maximum value of RCS and δ in the first kilometer above the surface.

2)    Three candidates are determined according to the maximum of $W_{RCS}$ and the maximum and minimum of $W_\delta$. In this sense, tThe first candidate (the so-called $C_{RCS}$) is determined is determined as the height of the maximum of $W_{RCS}$ closest to the surface exceeds a certain threshold $\eta_{RCS}$. This threshold is decreased iteratively, starting in 0.05, until $C_{RCS}$ is found. This is procedure established by Granados-Muñoz et al. (2012). A dilation value ($a_{RCS}$) of 300 m is used according to

Granados-Muñoz et al. (2012). Similarly, $C_{max}$ and $C_{min}$ are determined as the height of the maximum and minimum of $W_\delta$ closest to the surface exceeding the thresholds $\eta_{max}$ or $\eta_{min}$. Values of these thresholds will be determined during the





optimization process explained latter.$C_{min}$ and $C_{max}$ indicate are the heights of the strongest increase and decrease of δ, respectively.

3)   The $z_{PBL}$ attribution is performed comparing the relative location of the candidates since we have experimentally found that the height distribution of the candidates is linked to different aerosol layering. Fig. 3 shows an example of the $z_{PBL}$ attribution at 20:30 UTC on 16 June 2013.The normalized RCS and δ profiles at 532 nm are shown in left axis whereas their WCT ($W_{RCS}$ and $W_\delta$) are shown in right axis. The candidates $C_{RCS}$, $C_{max}$ and $C_{min}$ are shown in both axis.As can be seen, $C_{RCS}$ and $C_{max}$ are located at 5.2 km asl whereas $C_{min}$ is located around 1.3 km asl. We do not expect the $z_{PBL}$ around 5 km asl considering the experimental site latitudes and the hour of the day (At 20:30 UTC), and thus, $C_{RCS}$ and $C_{max}$ are probably detecting the top of an aerosol layer coupled to the PBL. However, $C_{min}$ shows an abrupt increase of δ caused by the transition between the lowermost and the coupled layers. This behavior is due to the lower depolarization capabilities of the anthropogenic aerosol, mainly presented within the PBL, in comparison with the mineral dust layer, coupled to the PBL. Thus, the abrupt increase of δ at $C_{min}$ is related to the top of the PBL,$z_{PBL}$, and thus, $C_{min}$ is chosen as $z_{PBL}$ instead of $C_{max}$ or $C_{RCS}$.

=>).

4)   Different scenarios are schematized in the flow chart shown in Fig. 4 and explained below:

a.   A candidate is not found: the $z_{PBL}$ corresponds to the minimum of the two others candidates (Fig. 4 and 5 case A).

b.   The three candidates are successfully determined: in this case, the attribution of the $z_{PBL}$ has two well-differentiated ways:

b.1.   Two matching candidates ($C_{CRS}=C_{max}$ or $C_{CRS}=C_{min}$): it is considered that $C_{CRS}$ matches $C_{max}$ or $C_{min}$ two candidates when the distance between them is less than 150 m. In these cases, the highest (in altitude) of the matching candidates is discarded, leaving only two candidates. Then, we define two layers: from the full-overlap height up to the lowest candidate, and from the lowest layer up to the highest candidate. Then, we retrieve the average and the variance of δ for both layers. When the absolute difference between the average value of δ is lower than a threshold $\delta_t$ and the variances differ less than 30%, the aerosol type in both layers are considered equal indicating that mixing processes evolve up to the highest candidate. Thus, the $z_{PBL}$ is attributed to the maximum of the two candidates(Fig. 4 and 5 case B or D).Conversely, if the aerosol type is different in both layers, there is not mixing between the layers and thus, the lowest candidate is the $z_{PBL}$(Fig. 4 and 5 case C or E).

b.2.   No match among the candidates: this situation indicates that the sharpest decrease of the RCS does not coincide with the sharpest decrease/increase of the δ.

b.2.1.   $C_{max}>C_{min}>C_{RCS}$: this situation is experimentally linked to an aerosol layer coupled to the PBL or a lofted aerosol layer within the free troposphere. In the case of aerosol layer coupled to the PBL, $C_{max}$ is the top of the coupled layer (i.e., $C_{max}$ is not the $z_{PBL}$); $C_{min}$ is the limit between the PBL and the coupled layer; and $C_{RCS}$ is an edge of an internal structure within the PBL. In the case of lofted aerosol





layer, $C_{max}$ and $C_{min}$ are the top and the base of a lofted layer, respectively whereas $C_{RCS}$ is the $z_{PBL}$. Since the top of a lofted layer would also show an increase of the RCS at the same altitude that $\delta$ increases ($C_{min}$), we search a local minimum of the $W_{RCS}$ around $C_{min}$ (i.e., $min\big(W_{RCS}(C_{min} \pm 50\,m)\big)$ larger than $\eta_{RCS}^{min}$). If found, $C_{min}$ is the bottom of a lofted layer and thus, the $z_{PBL}$ corresponds to $C_{RCS}$

5       (Fig. 4 and 5 case F). Otherwise, $C_{min}$ detects the $z_{PBL}$ (Fig. 4 and 5 case G).

b.2.2.   $C_{min}>C_{max}>C_{RCS}$: this situation indicates that RCS decreases at different altitudes than $\delta$ ($C_{max}>C_{RCS}$) with an increase of $\delta$ before $C_{max}$ and $C_{RCS}$. This situation is linked to a multi-layered PBL and thus, the attribution of the $z_{PBL}$ is performed looking for the altitude at which both RCS and $\delta$ profiles have the sharpest decrease. To this aim, $\Sigma_{max}$ and $\Sigma_{RCS}$ are defined by:

$$\Sigma_{max} = W_\delta(C_{max}) + max\big(W_{RCS}(C_{max} \pm 50\,m)\big) \hspace{3cm} \text{Eq. 2}$$

$$\Sigma_{RCS} = W_{RCS}(C_{RCS}) + max\big(W_\delta(C_{RCS} \pm 50\,m)\big) \hspace{3cm} \text{Eq. 3}$$

where $max\big(W_{RCS}(C_{max} \pm 50\,m)\big)$ is the maximum of $W_{RCS}$ in the range $C_{max} \pm 50\,m$ and $max\big(W_\delta(C_{RCS} \pm 50\,m)\big)$ is the maximum of $W_\delta$ in the range $C_{RCS} \pm 50\,m$. Physically, the parameters $\Sigma_{max}$ and $\Sigma_{RCS}$ are the sum of the WCT where both RCS and $\delta$ profiles have a sharp decrease. Then, if

$\Sigma_{max} > \Sigma_{RCS}$, both RCS and $\delta$ decrease at $C_{max}$ stronger than at $C_{RCS}$ and thus, the $z_{PBL}$ is attributed to $C_{max}$ (Fig. 4 and 5 case J), otherwise to $C_{RCS}$ (Fig. 4 and 5 case I).

b.2.3.   In the rest of combination of $C_{min}$, $C_{max}$ and $C_{RCS}$ not considered in b.2.1 and b.2.2, the $z_{PBL}$ is attributed to the minimum of the candidates ($C_{min}$ and $C_{max}$) (e.g., Fig. 4 and 5 case H).

Finally, the temporal coherence of the $z_{PBL}$ is checked using the procedure proposed by Angelini et al. (2009) and Wang et al. (2012). Once $z_{PBL}$ has beenis determined for a certain period, each $z_{PBL}$ is compared with its previous and subsequent value. According to the temporal evolution of the $z_{PBL}^{MWR}$, it is estimated that a difference between two variations with consecutive $z_{PBL}^{POL}$ the previous and subsequent values larger than 300 m is unrealistic. Thus, the $z_{PBL}^{POL}$ considered unrealistic is replaced by the average value of its three or six previous and latter values subject to availability. In this way we guarantee the

smoothness of the temporal series of the $z_{PBL}$. In addition, aerosol stratification could cause an inappropriate attribution of the $z_{PBL}$. However, as stratification presents short temporal duration compared to the mixing-layer temporal evolution (Angelini et al., 2009), a 7-bin moving median filter is used to reject the possible attributions related to aerosol stratification.

## 3.3 POLARIS adjustment

Fig. 6 shows the time series of the RCS and $\delta$ at 532 nm for the 36-hour lidar measurement (10:00 UTC 16 – 19:30 UTC 17

June) of ChArMEx 2013 campaign, the $C_{RCS}$, $C_{max}$ and $C_{min}$ candidates and the $z_{PBL}^{POL}$ and $z_{PBL}^{MWR}$ are shown in Fig. 6. This measurement was used to optimize the algorithm, the dilation $a_\delta$ and the different thresholds ($\eta_{min}$, $\eta_{RCS}$, $\eta_{RCS}^{min}$, and $\delta_t$). Following a similar procedure explained by Granados-Muñoz et al.(2012), different combinations of dilation and threshold



values were used to compute $z_{PBL}^{POL}$ and then compared to $z_{PBL}^{MWR}$ to establish the optimum values for the automatic detection of the PBL. Optimal $a_\delta$ is established at 450 m which is larger than the $a_{RCS}= 300$ m determined by Granados-Muñoz et al. (2012). This difference is because $\delta$ used to be noisier than RCS. The thresholds, $\eta_{min}$ and $\eta_{max}$ (used to find $C_{min}$ and $C_{max}$ by means of the minimum and maximum of $W_\delta$) are equal to $\eta_{RCS}$ (0.05) in absolute value. In the case of $\eta_{RCS}^{min}$, threshold used to distinguish decoupled layers, a value of 0.01 is chosen. Finally, the threshold $\delta_t$ (used in the case b.1) is established as 0.06 according to the results obtained in the optimization process.

During night-time, $C_{RCS}$ mainly almost does not detect the edges between the top of the PBL and the different stratifications within the dust layer, overlaying the PBL. However, POLARIS distinguishes the transition between the residual aerosol layer and the dust layer. The mean and standard deviation of the $C_{RCS}$ and the $z_{PBL}^{POL}$ is 3.1±1.6 and 1.5±0.3 km asl, respectively, for the period from 20:30 UTC on 16 June to 04:00 UTC 17 June. In comparison with $z_{PBL}^{MWR}$, $z_{PBL}^{POL}$ shows a more slow decreasing at the beginning of night and maintain an small ($\approx$ 300 m) offset with $z_{PBL}^{MWR}$ practically constant until the sunrise. Therefore, during night-time, POLARIS notably improves the detection of the $z_{PBL}$ during night-time since POLARIS provides lower $z_{PBL}$ than the method which uses only the RCS ($C_{RCS}$) and because POLARIS provides a better temporal behaviour.

During daytime, $z_{PBL}^{POL}$, $z_{PBL}^{MWR}$, and $C_{RCS}$ are compared. On 16 June 2013, the mean and standard deviation of $z_{PBL}^{POL}$, $z_{PBL}^{MWR}$ and $C_{RCS}$ is are 3.4±0.4, 2.7±0.3 and 2.2±1.1 km asl, respectively. $C_{RCS}$ is more than 1 km lower than $z_{PBL}^{MWR}$ probably because $C_{RCS}$ indicates layering points to a weak edge within the PBL. The large standard deviation of the $C_{RCS}$ is (1.1 km) is due to some detections around 1.8 km (weak edge within the PBL) and several ones at 4.5 km asl around 14:50 UTC (top of the dust layer) (Fig. 6). These results evidence that the $z_{PBL}$ detection fails using only the RCS profile but using POLARIS, when a dust layer is overlaying the PBL. Besides, $z_{PBL}^{POL}$ and $z_{PBL}^{MWR}$ is, in general, very similar although the $z_{PBL}^{POL}$ values are lower than the $z_{PBL}^{MWR}$ ones. Also, a delay of the $z_{PBL}^{POL}$ increase with respect to the $z_{PBL}^{MWR}$ increase during the transition from the residual layer to the mixing one. For example, $z_{PBL}^{POL}$ increases abruptly from 1200 to 2500 m asl between 11:20 and 11:30 UTC whereas $z_{PBL}^{MWR}$ increases from 1.48 km to 2.7 km between 10:15 and 11:30 UTC (i.e., almost one hour delay). These discrepancies could not be fixed during the optimization process due to their different basis: $z_{PBL}^{MWR}$ method uses thermodynamic variables as tracer whereas POLARIS uses the aerosol. Therefore, $z_{PBL}^{MWR}$ increases with the development of the convective processes but the vanishing of the residual layer edge (aerosol as tracer) only occur once the convection processes are strong enough. Besides discrepancies, both $z_{PBL}^{POL}$ and $z_{PBL}^{MWR}$, with low standard deviations, show comparable temporal evolution indicating the goodness of the method and thus, POLARIS also improves the $z_{PBL}$ detection during daytime.

## 4 Validation of POLARIS

After the optimization process, POLARIS is applied in an automatic and unsupervised way to the 72-hour lidar measurement performed during the ChArMEx 2012 campaign (between 9 and 12 July 2012). POLARIS is evaluated comparing $z_{PBL}^{POL}$, with $z_{PBL}^{MWR}$ and $z_{PBL}^{RCS}$. During this campaign, a Saharan dust outbreak occurred over the Southern Iberian Peninsula. As it can be



seen in Fig. 7, δ values are lower close to the surface (mainly local anthropogenic aerosols) in comparison with the lofted aerosol layers (dust aerosol plumes).

POLARIS and the method applied by Granados-Muñoz et al. (2012) ($C_{RCS}$) agree with discrepancies lower than 250 m when the dust layer is decoupled of the PBL (e.g., 00:00-08:00 UTC 10 July, 00:00-09:00 UTC 11 July and 18:00 11 July - 04:45 UTC 12 July). Therefore, the use of δ profiles is appropriate without coupled layers.

The comparison between $z_{PBL}^{POL}$ and $z_{PBL}^{MWR}$ shows a good agreement when the convection is well developed (13:00-16:00 UTC on each day). However, some discrepancies are found (e.g., 14:46 UTC 10 July 2012 and 15:51 UTC 11 July 2012). A detailed analysis of the temporal evolution of the RCS concludes that these differences are due to the high temporal fluctuation of $z_{PBL}^{MWR}$ likely associated to the high sensitivity of the parcel method to the surface temperature (e.g., small surface temperature variations may lead to large $z_{PBL}^{MWR}$ ones) and thus, it not related to fails in the POLARIS' performance (there is not vertical change of the aerosol load at $z_{PBL}^{MWR}$).

The large differences between $z_{PBL}^{POL}$ and $z_{PBL}^{MWR}$ occur during night-time (e.g. 20 UTC 9 July) due to the POLARIS detection of the residual layer whereas the $z_{PBL}^{MWR}$ indicates the stable layer between 100 and 300 m above ground level. At this range, MULHACEN is almost 'blind' since the overlap between the laser and telescope is too low, and thus, the lidar measurement has not information about the stable layer. Therefore, POLARIS detects the residual layer (i.e., the next edge above the stable layer). Despite POLARIS improves the detection at low altitudes by means of the δ profiles, the overlap region cannot be completely corrected and thus, the residual layer top will be detected instead of the stable layer when the stable layer is below the overlap height of the δ profiles. Furthermore, the comparison between $z_{PBL}^{POL}$ and $z_{PBL}^{MWR}$ revealed showed that the detection of the $z_{PBL}$ becomes particularly difficult when the mixing is ongoing (07:00-13:00 GMT) coexisting the residual and mixing layer. As it can be seen in Fig. 7 from 07:00 until 13:00 UTC on 11 July, $z_{PBL}^{MWR}$ is increasing (mixing layer is growing) whereas $z_{PBL}^{POL}$ is decreasing from 07:00 until 12:20 UTC (subsidence of the residual layer). During this period, despite the MWR points to convective processes, δ shows a layered structure. Therefore, the convective processes already initiated does not produces the necessary mixing that leads to the suppression of the residual layer. In fact, according to the δ edges provided by POLARIS (red and yellow triangles, Fig. 8), the mixture is almost complete around 13:15 UTC.

## 5. WRF validation using POLARIS and MWR

Recent studies uses the $z_{PBL}$ determined using lidar data to validate the $z_{PBL}$ obtained from WRF model ($z_{PBL}^{WRF}$) (Xie et al., 2012; Pichelli et al., 2014 and Banks et al., 2015). In this section, we take the advantage of the $z_{PBL}$ determined by POLARIS ($z_{PBL}^{POL}$) together with the microwave radiometer $z_{PBL}^{MWR}$ during CHArMEx 2012 and 2013 to validate the $z_{PBL}^{WRF}$ under complex atmospheric conditions.

### 5.1 WRF model setup





The WRF NWP model, version 3.6.1, was used to analyse the CHARMEX 2012 and 2013 campaigns. The model configuration consists of four nested domains with 27, 9, 3 and 1 km (approximately) spatial resolution domains, respectively, and 50 vertical levels. The outputs (i.e., temperature, wind, and humidity profiles, etc.) of the 1-km domain were analysed. The initial and boundary conditions for the WRF model runs are taken from the NCEP High Resolution Global Forecast System data set (www.emc.ncep.noaa.gov) every 6 hours. The 1-km WRF outputs are saved every 5 minutes.

The choice of the model physical parameterization was based on the results of previous evaluation studies conducted in the study area (Arbizu-Barrena et al., 2015; Santos-Alamillos et al., 2013). Particularly, the Mellor-Yamada Nakanishi and Niino Level 2.5 was selected for the PBL parameterization (Nakanishi and Niino, 2009). This parameterization performs Turbulent Kinetic Energy advection and accounts for both sensible and latent heat fluxes as well as moisture flux from the surface. The parameterizations used for the rest of physical schemes are: the Eta (Ferrier) microphysics parameterization scheme (Rogers

et al., 2005), the RRTM long-wave radiation parameterization (Mlawer et al., 1997), the Dudhia scheme for short-wave radiation parameterization (Dudhia, 1989), the 5-layer thermal diffusion land surface parameterization (Dudhia, 1996) and, for coarser domains, the Kain-Fritsch (new Eta) cumulus parameterization (Kain, 2004).

**5.2 Comparison of the PBL heights determined by WRF, POLARIS and microwave radiometer**

Fig. 6 and 7 shows the temporal evolution of the PBL heights determined by means of POLARIS, the MWR and WRF, $z_{PBL}^{POL}$, $z_{PBL}^{MWR}$, and $z_{PBL}^{WRF}$, respectively. The period represent accounts for the RCS and $\delta$ at 532 nm during the ChArMEx campaign on 2012 (09:00 UTC 16 June– 20:00 UTC 17 June) and 2013 (12:00 UTC 9 July – 06:00 12 July).

During daytime on both campaigns, WRF underestimates the $z_{PBL}$ (lower values) with respect to $z_{PBL}^{POL}$ and $z_{PBL}^{MWR}$ in agreement

with the study presented by Banks et al. 2015 and Banks and Baldasano, 2016. For example, $z_{PBL}^{WRF}$ is 1 km below $z_{PBL}^{POL}$ and $z_{PBL}^{MWR}$ on 16/06 2013 (Fig. 6) and on 9 and 10 July 2012 (Fig. 7). Nevertheless, the $z_{PBL}$ time series of all methods show similar patterns Table 1 shows the correlation factor $R^2$ and the mean of the differences (i.e., bias) among $z_{PBL}^{WRF}$, $z_{PBL}^{POL}$ and $z_{PBL}^{MWR}$. Correlations between $z_{PBL}^{WRF}$ and $z_{PBL}^{POL}$ ($R^2_{POL-WRF}$) and between $z_{PBL}^{WRF}$ and $z_{PBL}^{MWR}$ ($R^2_{MWR-WRF}$) are usually greater than 0.6 during daytime period. Particularly, on 10 July during ChArMEx 2012, the correlation is very good ($R^2_{POL-WRF}$ and

$R^2_{MWR-WRF}$ are 0.763 and 0.605, respectively), but with $\overline{\Delta_{PBL}^{POL-WRF}}$ and $\overline{\Delta_{PBL}^{MWR-WRF}}$ of 679 and 411 m, respectively. A good correlation ($R^2_{POL-WRF}$=0.661) with a large bias ($\Delta_{PBL}^{POL-WRF}$ = 1171 m) between is also detected on 11 July 2012 during daytime. These deviations can be associated to the WRF performance, pointing to an underestimation of the convective processes. At this regard, several possibilities are feasible. (i) Too stringent conditions for the WRF parameterization, which can influence directly the results (Xie et al., 2012; Banks et al., 2015). (ii) Insufficient number of the WRF model vertical levels near the

PBL limits. (iii) The Saharan dust layer strongly coupled to the PBL (see $\delta$ temporal series on 9 July 2012 and on 17 June 2013 in Fig. 6 and 7). (iv) Different definition of PBL applied to each method (Xie et al., 2012). From these causes, the first and second ones may affect to the whole period and the differences are too large to be caused by the possible different definitions of PBL (iv). In fact, POLARIS and the parcel method use different tracers for the PBL height detection (e.g.,



temperature, MWR, and aerosol, POLARIS) with similar results. Thus, probably the most probable cause is the presence of the Saharan dust layer strongly coupled to the PBL, not properly accounted by WRF.

The lowest correlations between POLARIS and WRF occur on 16 and 17 June 2013. On 16 June, the correlations between POLARIS and WRF ($R^2_{POL-WRF} = 0.122$) and between MWR and WRF ($R^2_{MWR-WRF} = 0.395$) are very low with respect to the

correlation between POLARIS ($R^2_{POL-MWR} = 0.803$). On this day, The period with convective processes is between 13:35 and 16:15 UTC for the WRF model whereas the MWR detects convective processes between 10h30 and 18h00 (i.e, 5 hours of difference). Therefore, the short duration of the convective processes estimated by the WRF model seems to be the cause of these differences found on 16 June 2013. On 17 June, the low correlation between POLARIS and WRF coincide with the lowest bias determined among POLARIS, MWR and WRF ($\overline{\Delta^{POL-WRF}_{PBL}}$ and $\overline{\Delta^{MWR-WRF}_{PBL}}$ of 236 and 275 m). The presence of

clouds from midday (cloud base at 10 km asl) until the end of the measurements (cloud base at 2 km asl may explain this behaviour since i) the systematic underestimation from WRF would be compensated by the cloudy conditions inhibiting the strength of convective processes and ii) the track of the complex temporal evolution of the PBL during cloudy conditions is more difficult to fit considering the different tracers (i.e., aerosol and temperature).

During night-time, $z^{WRF}_{PBL}$ and $z^{MWR}_{PBL}$ closely agree, with differences below 200 m (see Table 1) and even being the same during

some periods (e.g., from 01:52 to 05:11 UTC on 10 July 2012, see Fig 8) whereas they show almost any correlation with $\overline{\Delta^{MWR-WRF}_{PBL}}$ values between 0.032 and 0.364. This is the opposite behaviour between MWR and WRF than during daytime. The large bias $\overline{\Delta^{POL-WRF}_{PBL}}$ and $\overline{\Delta^{POL-MWR}_{PBL}}$ evidence that the stable layer height is generally too low to be detected by POLARIS, and thus, POLARIS provides the top of the residual layer. Note that, overall, the $R^2_{POL-WRF}$ and $R^2_{MWR-WRF}$ values point to a more similar behaviour between POLARIS and WFR than between MWR and WRF.

To sum up, during daytime, the WRF model underestimates the $z_{PBL}$, but the temporal evolution closely agrees with that of the $z_{PBL}$ experimentally determined. During night-time, values reported by the WRF closely agree with the experimental MWR $z_{PBL}$ values.

## 6. Conclusion

The perpendicular-to-parallel signal ratio (i.e., the uncalibrated volume linear depolarization ratio), together with the lidar range corrected signal, is used to develop a new algorithm, called POLARIS, for the detection of the planetary boundary layer height ($z_{PBL}$). Firstly, the $z_{PBL}$ provided by POLARIS, $z^{POL}_{PBL}$, has been optimized by comparison with the $z_{PBL}$ derived from microwave radiometer measurements (temperature profiles), $z^{MWR}_{PBL}$, using continuous 36-hour lidar and MWR measurements. Secondly, $z^{POL}_{PBL}$ has been validated by comparison with the $z^{MWR}_{PBL}$, using a using continuous 72-hour lidar and MWR

measurements. These measurements were performed in the ChArMEx campaigns conducted in 2012 (36-hour) and 2013 (72-hour). These long-term measurements have been crucial for the adjustment and validation since it allows the tracking of the evolution of the coupling between the advected aerosol layers and the planetary boundary layer. A good agreement between $z^{POL}_{PBL}$ and $z^{MWR}_{PBL}$ have been obtained even when a Saharan dust layer was coupled to the PBL. This is an important advance





since the false detections produced by advected layers coupled to the PBL used to be large as it was evidenced comparing the $z_{PBL}$ derived from the RCS and from POLARIS. Moreover, considering the next ceilometer generations with depolarization capabilities, POLARIS will be useful for an automatic and unsupervised PBL detection.

The $z_{PBL}$ has been also determined by means of WRF model, $z_{PBL}^{WRF}$. During daytime, $z_{PBL}^{WRF}$ were considerably lower than $z_{PBL}^{POL}$

and $z_{PBL}^{MWR}$ with larger differences under coupling-layer situation. However, WRF and MWR provides similar $z_{PBL}$ during night-time although $z_{PBL}^{WRF}$ shows a better correlation with $z_{PBL}^{POL}$ than with $z_{PBL}^{MWR}$. The comparison between POLARIS and WRF evidences the difficulties of the models to determine the $z_{PBL}$ when advected layers are coupled to the PBL. Therefore, POLARIS allows a better model validation since it provides confident PBL heights event under complex atmospheric situations. Further investigations in this regards would lead to a proper PBL height detection in all atmospheric conditions.

**Acknowledgements**

This work was supported by the Andalusia Regional Government through project P12-RNM-2409, by the Spanish Ministry of Economy and Competitiveness through project CGL2013-45410-R and by the European Union's Horizon 2020 research and innovation programme through project ACTRIS-2 (grant agreement No 654109). The authors thankfully acknowledge the FEDER program for the instrumentation used in this work. This work was also partially funded by the University of Granada

through the contract "Plan Propio. Programa 9. Convocatoria 2013". The authors express gratitude to the ChArMEx project of the MISTRALS (Mediterranean Integrated Studies at Regional and Local Scales; http://www.mistrals-home.org) multidisciplinary research programme.

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




Table 1:R² among $z_{PBL}^{POL}$, $z_{PBL}^{MWR}$ and $z_{PBL}^{WRF}$ during ChArMEx 2012 and 2013. Points are the number of values used to retrieve the correlation factor. $\overline{\Delta_{PBL}^{POL-WRF}}$, $\overline{\Delta_{PBL}^{MWR-WRF}}$, $\overline{\Delta_{PBL}^{POL-MWR}}$ is the mean difference between the $z_{PBL}^{POL}$ and $z_{PBL}^{WRF}$, $z_{PBL}^{MWR}$ and $z_{PBL}^{WRF}$ and $z_{PBL}^{POL}$ and $z_{PBL}^{MWR}$, respectively .Daytime is considered between 06:00 and 19:00 UTC and night-time is the rest of the day.

| | | Daytime | $R^2_{POL-WRF}$ | Points | $\overline{\Delta_{PBL}^{POL-WRF}}$(m) | $R^2_{MWR-WRF}$ | Points | $\overline{\Delta_{PBL}^{MWR-WRF}}$(m) | $R^2_{POL-MWR}$ | Points | $\overline{\Delta_{PBL}^{POL-MWR}}$(m) |
|---|---|---|---|---|---|---|---|---|---|---|---|
| ChArMEx | 2012 | 9th July | 0.236 | 12 | 850 | 0.664 | 12 | 438 | 0.598 | 12 | 376 |
| | | 10th July | 0.763 | 26 | 679 | 0.605 | 26 | 411 | 0.718 | 26 | 244 |
| | | 11th July | 0.661 | 26 | 1171 | 0.441 | 26 | 520 | 0.361 | 26 | 1697 |
| | 2013 | 16th June | 0.122 | 26 | 826 | 0.395 | 26 | 1333 | 0.803 | 26 | 572 |
| | | 17th July | 0.018 | 26 | 2323 | 0.094 | 26 | 275 | 0.304 | 26 | 42 |
| | | Night-time | $R^2_{POL-WRF}$ | Points | $\overline{\Delta_{PBL}^{POL-WRF}}$(m) | $R^2_{MWR-WRF}$ | Points | $\overline{\Delta_{PBL}^{MWR-WRF}}$(m) | $R^2_{POL-MWR}$ | Points | $\overline{\Delta_{PBL}^{POL-MWR}}$(m) |
| | 2012 | 9th July | 0.660 | 28 | 938 | 0.364 | 17 | 185 | 0.463 | 17 | 1154 |
| | | 10th July | 0.640 | 28 | 930 | 0.032 | 9 | 175 | 0.057 | 9 | 1126 |
| | | 11th July | 0.440 | 28 | 767 | 0.230 | 11 | 380 | 0.062 | 11 | 1130 |
| | 2013 | 16th June | 0.030 | 28 | 388 | 0.099 | 9 | 400 | 0.028 | 9 | 733 |
| | | 17th July | - | - | | - | - | | - | - | |

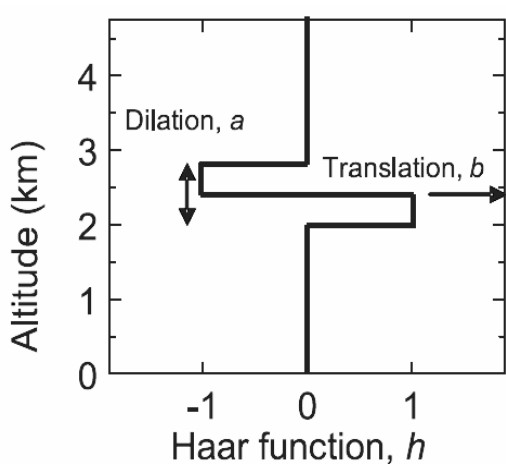

Figure 1: Haar's function defined by the dilation ($a$) and the translation ($b$).





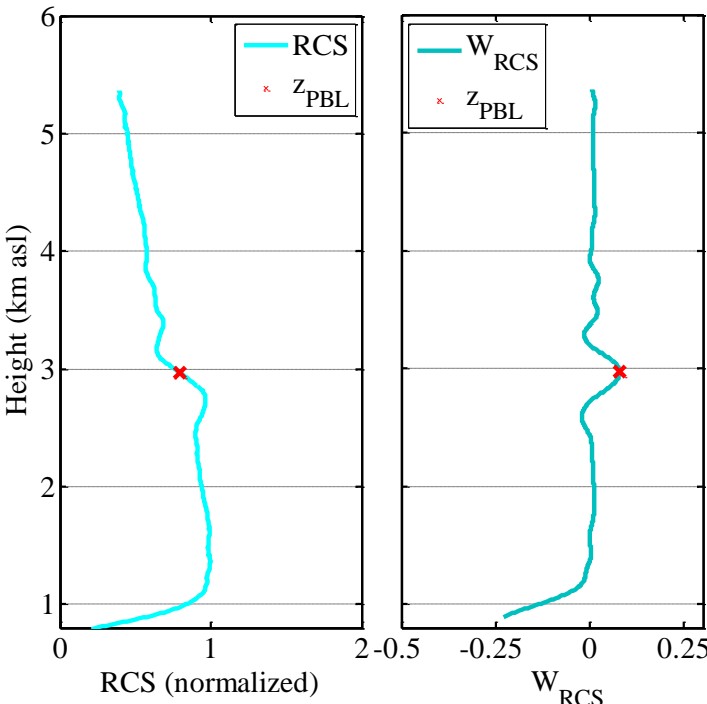

**Figure 2: Example of a normalized RCS and its wavelet covariance transform. Red cross indicates the possible location of the PBL height.**





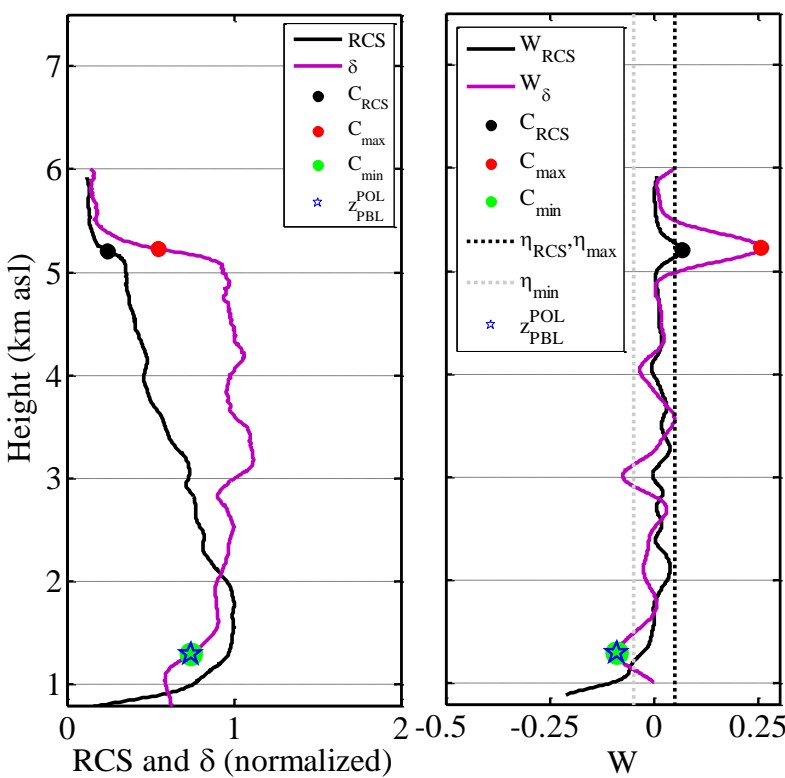

**Figure 3: Normalized RCS and δ profiles (left). WCT of the RCS, δ and thresholds $\eta_{min}$ (−0.05) and $\eta_{RCS}$, $\eta_{max}$ (0.05) (right) at 20:30 UTC 16 June 2013. $C_{CRS}$, $C_{min}$ and $C_{max}$ candidates and $z_{PBL}^{POL}$ are shown in both axes.**





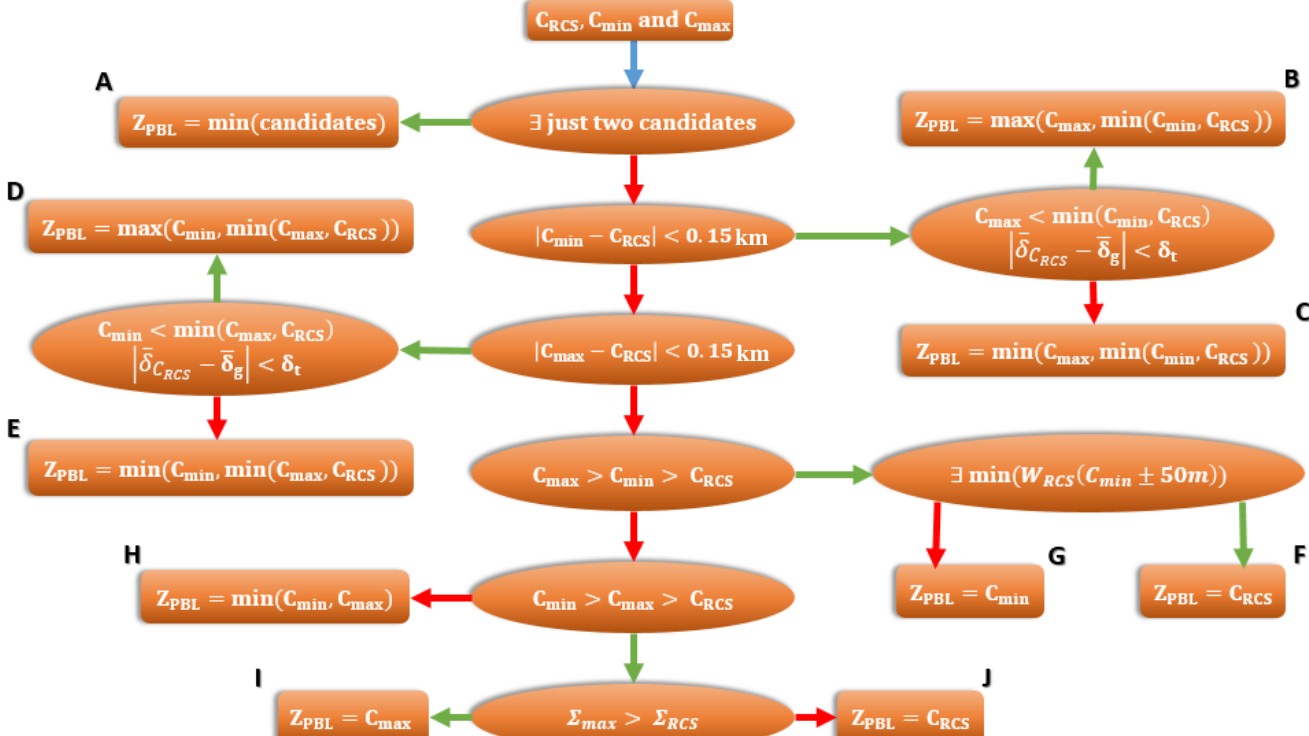

**Figure 4:** Flux diagram of the algorithm used by POLARIS to determine the $z_{PBL}$. $C_{min}$, $C_{max}$ and $C_{RCS}$ are the candidates. The blue arrow indicates the start. Conditions are marked in ellipses and the final attribution of the $z_{PBL}$ in rectangles. The green and red arrows indicate the compliance and noncompliance of the conditions, respectively. The rest of the symbols are explained in the text.







**Figure 5: Examples of the cases mentioned in Figure 4 that occurred during ChArMEx 2012 and ChArMEx 2013.**
Normalized RCS (dark red line) and $\delta$ (purple line) are shown in left axis and WCT of RCS (grey line) and $\delta$ (red line) are shown in right axis. $C_{min}$(green dot), $C_{max}$(red dot), $C_{RCS}$ (black dot) and the final attribution $z_{PBL}^{POL}$(blue start) are shown in both axis.



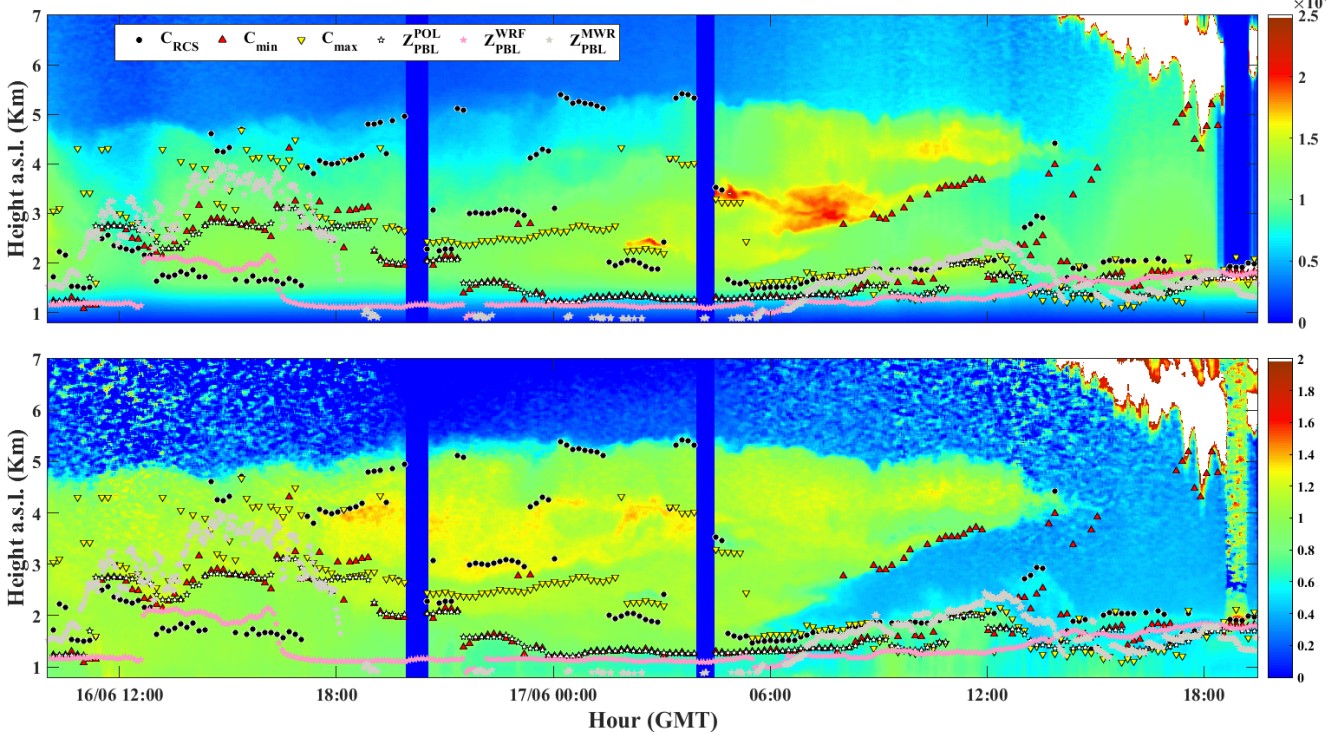

**Figure 6: Temporal evolution of the Range Corrected Signal (RCS) and the perpendicular-to-parallel signal ratio (δ) in the period 09:00 16 June - 20:00 17 June 2013 (colour maps). The scatter plots represent the candidate for $Z_{PBL}$($C_{RCS}$ (black dot), $C_{min}$ (red triangle) and $C_{max}$ (yellow inverted triangle)). The purple stars and the pink dots are the $Z_{PBL}$ determined with POLARIS and the parcel method using MWR measurements, respectively. Measure gaps are dark-current measurements**





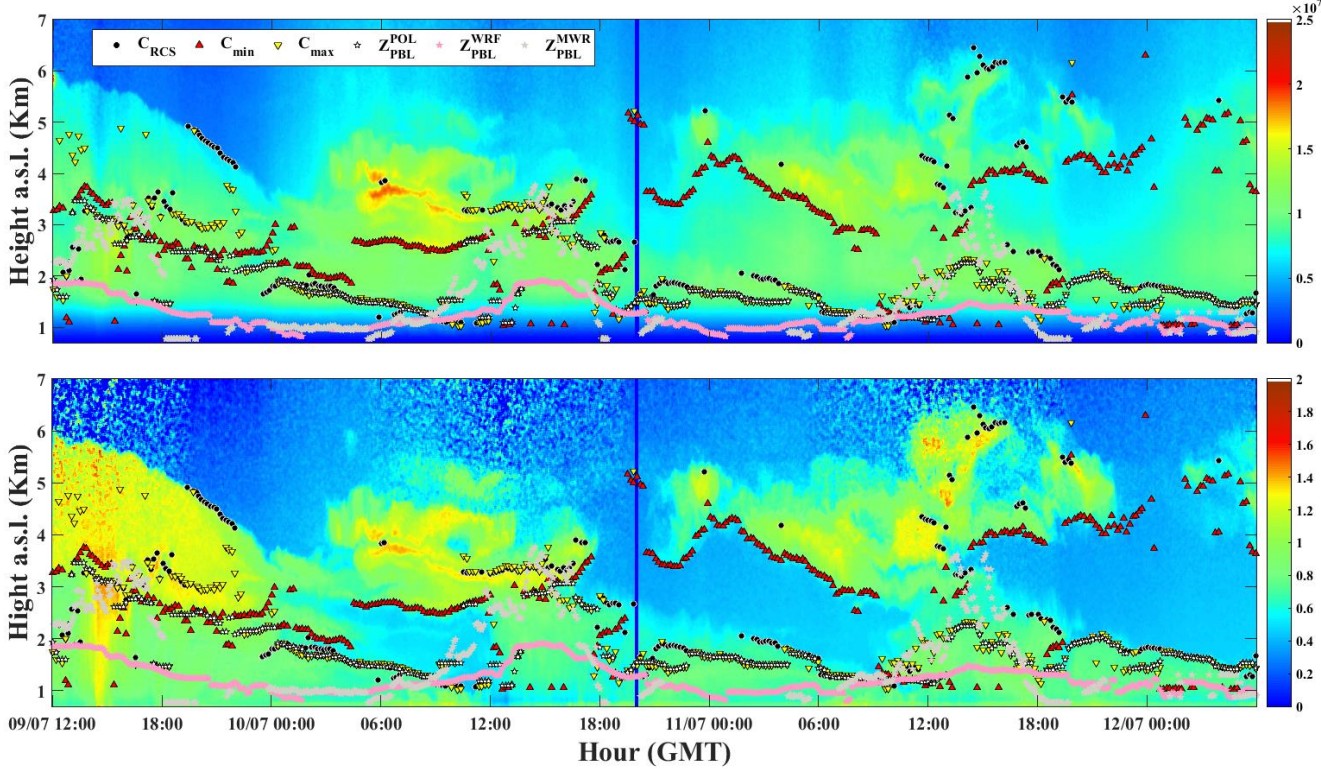

Figure 7: RCS and δ temporal evolution in the period 12:00 9 July – 06:00 12 July 2012 (color maps). Purple stars and pink dots represent $z_{PBL}^{POL}$ and $z_{PBL}^{MWR}$, respectively.





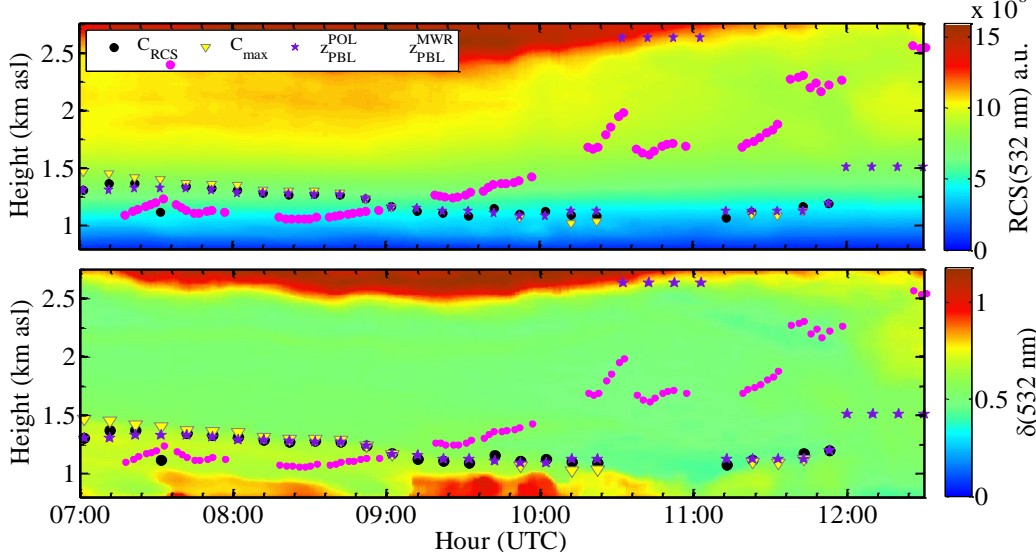

**Figure 8: Zoom of Fig. 7 showing the RCS and δ temporal evolution during the period 07:00-13:00 UTC on 10 July 2012 (colour maps). C$_{RCS}$ (black dots), C$_{max}$ (yellow triangles), $z_{PBL}^{POL}$ (purple stars) and $z_{PBL}^{MWR}$ (pink dots) are icluded.**