# Peer review of "A new methodology for PBL height estimations based on lidar depolarisation measurements: analysis and comparison against MWR and WRF model based results"

_Atmospheric Chemistry and Physics, 2016_

## Referee Comment (RC1) · Anonymous Referee #1 · 12 Dec 2016

Review: The Paper by Bravo-Aranda et al. presents an advanced methodology based on existing work to estimate the height of the PBL top by aerosol lidar. As a novelty, the authors use the profile of the depolarization profile to better identify the mixing layer under conditions of complex aerosol layering. The paper could be suited for ACP after some revisions.

General comments:

-First of all, I am a bit angry because within the manuscript there are a lot of minor spelling mistakes which easily could have been found if a spelling checker would have been used or a co-author would have read the final version. Thus the authors should carefully read their manuscript again and remove these mistakes – I will not mark them separately. E.g., many "spaces" are missing and very often two dots at the end of a

sentence are found.

-Secondly, I think the statement made for the improvement of the PBL height is to strong. The methodology sounds very interesting, but is in my opinion limited to situations with Saharan dust intrusions as shown in the 2 case studies of Charmex. Therefore, general statements should be avoided. Instead it should be written that for the meteorological conditions like in Granada this methodology could be a significant improvement. As no long-term data set (>12 months is presented), one can only speculate that the new methodology is a significant improvement. Therefore some statements should be weakened. E.g: the statement in the introduction: P3,l3: "POLARIS improves the zPBL detection since the computation of . . ." is only valid, IF different aerosol with different depolarization characteristics are existing within and above the PBL. This is certainly not true for many sites not influenced by dust. Or: P5: line 19:" . . .able to detect the PBL height even when advected aerosol layers in the free troposphere are coupled to the PBL." Only if the advected aerosol layer show a different depolarization ratio.

-Also the quality of Figures 6 and 7 must be improved. With the current state the discussion is hard to follow. Symbols should be revised for better readability, time scale should have more tick marks, height scale should be probably revised. Often symbols lay very close to each other and have similar color. Reducing symbol frequency could be also an option to consider. Please do definitely choose a different symbol for the PBL top height with POLARIS. The star is not visible and all other symbols are much more prominent.

-Even so referred to a previous paper I miss a real discussion on the MWR PBL height trustworthiness. In this paper it is always taken as the truth and one wonders why to use the lidar at all. . .

- An lidar instrument discussion is needed: How trustworthy are profiles close to the lidar (overlap issue but also polarization properties).

- Title: I think the title is not representative for the paper as it not only deals with PO-LARIS. Probably, the model comparison approaches and MWR should be accounted for in the title.

-Section 5 should be shortened: Please only show up with new information an restrict to WRF. Almost no new information with respect to the other already published results of R. Banks are given. Probably use WRF to check consistency of retrievals (begin of convection etc.,)

Specific comments (no spelling mistakes):

Page 5, line 11. Unit missing after 0.05 and later in the manuscript. Or what do you mean with dilation parameter?

Page 5, line 21: What do you mean: can be applied to lidars not fully characterized? I think this is a very "dangerous" statement

Page 6: line 12: how can you assure, that this increase is not due to instrumental effects? The ratio of two signals very close to the lidar might not cancel out all instrumental effects anymore, especially regarding depolarization. Do you have a case study where it is seen that the depolarization ratio is constant throughout the atmosphere? I always see an increase towards ground below 1 km.

Page 6, line: I did not understand what you mean with lowest layer, please rephrase: "Then, we define two layers: from the full-overlap height up to the lowest candidate, and from the lowest layer up to the highest candidate."

Page 6, line 26: I do not know what you mean with aerosol stratification in this context, can you explain better?

Page 7, line 4 to 6. Are these thresholds really without dimension?

Page 8, line 11: An offset of 300 m is not too small, how reliable are the MWR measurements during night time? And how reliable is your lowest candidate, as it is very

close to the lidar. I wonder if the depolarization measurements are reliable (see comment above), please discuss this. Especially as you state that Polaris improves the detection. I cannot follow this discussion, as I do not know the "truth".

Page 8, line 22. Between 11:20 and 11:30 UTC on which day? Again it is hard to follow, because the symbols are so tiny and especially the Polaris an (star) is almost invisible.

Page 8: Discrepancies between MWR and Polaris on 16 June afternoon should be more intensively discussed. I see a systematic bias of almost 1 km.

Page 9, line 22ff: Does the MWR really watch the same atmospheric column or is scanning included? Then the discussion could be different. Furthermore, also the overlap issue and the near-range depolarization issues could play a role.

Page 11 line 31: These are certainly no long-term measurements, use better continuous measurements

Page 12, line 7ff: POLARIS allows better model validation compared to what?

Fig. 5 please move legend to case B and enlarge - it is not readable even at 150% zoom. Sometimes the blue star changes the color? If the star is overlaid with a dot pleas make it visible.

Fig. 6 and 7. See above: Poor quality, work on improvement, otherwise discussion cannot be followed. Explanation for WRF missing in caption.

Fig. 8: pink dot is somewhere else in legend.

---

## Referee Comment (RC2) · Anonymous Referee #2 · 14 Dec 2016

I have not read the review of anonymous reviewer #1 in detail to avoid any "influence" from his/her criticism, but I fully agree that the quality of the presentation is unacceptable. It seems that none of the 10(!) authors was willing to provide a text fulfilling minimum formal standards of a manuscript. This is downright annoying and disrespectful to their readership. From this point of view I would like to reject the paper.

On the other hand the consideration of lidar depolarization measurements can indeed be a promising approach to improve the retrieval of the mixing layer height. In their paper Bravo-Aranda et al. propose an extension (POLARIS) of an existing method based on the analysis of the range corrected lidar signal. The new approach is compared to the previous one (and it is found that it performs better), and to an independent retrieval utilizing data from a microwave radiometer (and it is found that it does not perform well, cf. doi:10.5194/amt-7-3685-2014). Finally, the mixing layer heights (MLH) derived from

measurements (lidar, MWR) are used to validate WRF-simulations.

The benefit of such an investigation certainly could be high, especially in view of future applications for ceilometers with a depolarization channel, in particular as these instruments are expected to have a lower overlap, can run unattendedly and continuously, and might be available as networks (doi:10.5194/amt-7-1979-2014). This is in my view the only argument not to reject the paper.

The present version of Bravo-Aranda's paper, however, suffers from several shortcomings. First, the small set of measurements is problematic. As a consequence, it cannot be demonstrated in a convincing way that POLARIS is useful when long time series shall be evaluated. Even for the short period of only 3 days (according to doi:10.5194/acp-16-455-2016, the SOP I of ChArMEx 2013 took place from 11. June to 5. July: what about this data-set?) a lot of situations were found when the different methodologies disagree. It is also not clear whether the basic assumption (if I understand it correctly, unfortunately it is not sufficiently explained) "a depolarizing aerosol layer is a transported desert dust layer and does not belong to the mixing layer" can be applied to other sites than Granada (or other Mediterranean countries). So this paper (provided the text has been undergone a substantial revision in grammar and spelling, and has been improved in terms of scientific clarity) can only be considered as a first contribution to a discussion of the benefit of adding depolarization-information to a MLH-retrieval.

I will not list the countless typos, word repetitions, cases of wrong grammar, misuse of capital letters, misspelled units, or undefined symbols. Even one of the affiliations is not correct and the link to the ChArMEx-Website does not work! Only a few specific mostly science-related issues are listed below.

In summary I think the authors should take all comments seriously. If not all issues are fully resolved I will not recommend the publication of a revised version.

Specific comments

[Figure]

- page 2, lines 14ff: There is an extensive discussion on different regimes as the residual layer, the mixing layer, the convective boundary layer and more. In the rest of the paper primarily the "PBL" is mentioned and discussed (see 3/5). A strict terminology is required throughout the paper. When there is a co-existence of the residual layer and the convective layer (e.g. after sunrise) PBL might be confusing.

- 2/23: "*Sunrise and Sunset are characterized by the complexity of the PBL.*": This sentence is really strange!

- 3/18: "*... are feasible and reliable ...*": What is meant with "reliable"? In the paper many example are shown when this is not the case.

- 3/21: "*... include stringent conditions ...*": What is this?

- 4/8: There are a lot of words on the overlap, but the most relevant number, i.e. the minimum range that can be exploited in terms of MLH, is missing. Why is the 90% overlap-range given as a interval? Is it temperature dependent? If it depends on the channel (i.e., wavelength) but only one wavelength is used in this study, it is not adequate to give a range.

  It is strange that the polarization channels which are the most relevant in view of the novelty of POLARIS are not mentioned here – whereas the irrelevant Raman-channels and water vapor channels are mentioned.

- 4/28: The vertical resolution given here does not agree with the statement in line 21.

- 5/24: "*both ... are normalized respectively to the maximum value of RCS and $\delta$ in the first kilometer above the surface*". In case of $\delta$ no normalization can be seen in any of the corresponding figures. Please clarify.

- section 3: It is not common to use the character "C" for a height.

- 6/4: Fig. 6 is discussed prior to Figs. 4 and 5. Fig. 3 is missing!

- 6/5: What is the reason for selecting RCS at 532 nm?

  Furthermore, "height above mean sea level" should be transformed to "height above ground" (throughout the paper).

- 6/8: "*We do not expect the ...*": Does this mean that an automated POLARIS retrieval is not possible?

- 6/15ff: The following discussion is confusing. A few examples: Under "b.1" it is stated that $C_{CRS} = C_{max}$ (by the way another typo: should be $C_{RCS}$) whereas in the next line of text the authors describe that $C_{CRS} \neq C_{max}$! It is not clear, what the "lowest layer" is (line 22). It is doubtful that at the top of a lofted layer RCS increases (7/2), the opposite should be the case. It is not clear why there is an increase of $\delta$ "before $C_{max}$" (7/7). It should be clearly outlined what should be understood by "coupled", it seems that it is used in different ways.

  It is difficult to understand a situation when $C_{min} > C_{max}$, whereas the opposite can be identified as e.g. a lofted dust layer.

- In Fig. 5 the differences of the profiles in cases D/E or F/G are hardly visible. Nevertheless the retrieval results in quite different $z_{PBL}$. This seems to be a weakness of the method and should be discussed in detail. Moreover, case I seems to be critical. The "inhomogeneities" in the shape of the $\delta$-profiles are not much pronounced so it seems questionable if depolarization should be exploited at all, especially when considering measurement errors (error bars are missing in all figures!).

  The labels of the axes and the legend are hardly readable.

- Section 3.3: A discussion of how the different thresholds are found is missing. There are only statements on specific numbers. The rest of the text does not

really fit to the title of the section; it is rather a discussion of the differences of the old and new method.

- 8/16: "$C_{RCS}$ *indicates layeringpoints to a weak edge within the PBL.*" Another example of a "weird" sentence.

- 8/21: Why do the authors switch to "m" instead of "km" as in the rest of the text?

- Section 4: Validation is performed by means of the MWR-retrieval. This implies that the latter is assumed to be the truth (see analysis in doi:10.5194/amt-7-3685-2014). As a consequence the MWR-retrieval and its accuracy has to be explained in more detail.

  In Section 4 the authors demonstrate that there are a lot of differences. Thus, the reader might conclude that the POLARIS-retrieval does not work reliably (in my view the grey and black stars never coincide in Fig. 7).

  In Fig. 7 it is not explicitly explained which parameter is shown in the upper/lower panel.

- 9/24: There are no red triangles in Fig. 9!

- Section 5: Obviously there are very few cases when POLARIS-retrievals agree with the WRF-simulations. What is the conclusion with respect to the usefulness of POLARIS or the accuracy of WRF?

---

## Author Comment (AC1) · 27 Mar 2017

The response to Reviewer #1 and Reviewer #2 is included in the supplement file.

Please also note the supplement to this comment:
http://www.atmos-chem-phys-discuss.net/acp-2016-718/acp-2016-718-AC1-supplement.pdf

---

## Author Comment (AC2) · 27 Mar 2017

acp-2016-718:

Reply to all comments

The authors would like to thank the reviewers for their thoughtful and helpful comments and suggestions. Their reviews have made a significant contribution to the improvement of the paper. The line numbering in the reviewers' comments refers to the manuscript published in ACPD whereas the line numbering in the responses refers to the new version of the manuscript.

Answer to general comments of both Anonymous Referees:

First of all, as first author, I have to apologize to the referees and the co-authors because I submitted an old version of the manuscript to the ACPD. I submitted the manuscript under pressure and thus I did not realized of my mistake. I would like to clarify that it was completely my fault and nothing related to the co-authors who have widely contribute to this work. I would like to thank the referees for paying attention to the scientific content of the manuscript despite the problems and for giving us the opportunity of replying.

The questions and comments from the Referees (in blue) are answered below (in green).

Answers to Anonymous Referee #1:

General comments:

Secondly, I think the statement made for the improvement of the PBL height is to strong. The methodology sounds very interesting, but is in my opinion limited to situations with Saharan dust intrusions as shown in the 2 case studies of Charmex. Therefore, general statements should be avoided. Instead it should be written that for the meteorological conditions like in Granada this methodology could be a significant improvement. As no long-term data set (>12 months is presented), one can only speculate that the new methodology is a significant improvement. Therefore some statements should be weakened. E.g: the statement in the introduction: P3,l3: "POLARIS improves the zPBL detection since the computation of..." is only valid, IF different aerosol with different depolarization characteristics are existing within and above the PBL. This is certainly not true for many sites not influenced by dust. Or: P5: line 19:"...able to detect the PBL height even when advected aerosol layers in the free troposphere are coupled to the PBL." Only if the advected aerosol layer show a different depolarization ratio.

This methodology can be applied in all stations affected by dust outbreaks which includes, at least, all the Mediterranean countries. This methodology is theoretically valid if different aerosol with different depolarization characteristics are present within and above the PBL. However, in the present study, it has been validated only for dust cases and further analysis with different aerosol types is needed to quantify the improvement. Taking into account this consideration, we have revised the whole document removing the strongest sentences.

-Also the quality of Figures 6 and 7 must be improved. With the current state the discussion is hard to follow. Symbols should be revised for better readability, time scale should have more tick marks, height scale should be probably revised. Often symbols lay very close to each other and have similar color. Reducing symbol frequency could be also an option to consider. Please do definitely choose a different symbol for the PBL top height with POLARIS. The star is not visible and all other symbols are much more prominent.

Figure has been updated following the suggestions of the Anonymous Referees #1 and #2.

-Even so referred to a previous paper I miss a real discussion on the MWR PBL height trust worthiness. In this paper it is always taken as the truth and one wonders why to use the lidar at all...

In this study, the optimization and validation of a new methodology to determine the PBL height has to be performed against a PBL height derived from independent measurements. Thus, we use the PBL height derived from MWR temperature profiles. We have modified the manuscript to highlight that the reference PBL includes also uncertainties and weak points in the methodology.

- An lidar instrument discussion is needed: How trustworthy are profiles close to the lidar (overlap issue but also polarization properties).

The following discussion has been included except the figure: '*The optical path of the parallel and perpendicular channels at 532 nm are designed to be identical up to the PBC where the 532 nm signal is split into parallel and perpendicular before reaching the PMT. This setup allows us to assume almost the same overlap for both polarizing components. Thus, the depolarization profile is practically not influenced by the incomplete overlap since it is cancelled out by the ratio of the perpendicular and parallel channels. Only the thermal dilation and contraction of the lidar optics*

[Figure]

Figure 1: MULHACEN overlap function at 532 nm. Extracted from Rogelj et al., 2014. Station height 680 m asl.

*after the PBC might independently change the overlap function of each channel. Since MULHACEN is deployed inside an air-conditioned building, the temperature fluctuation is small and thus, the overlap difference between the channels might be low. Therefore, we assume significant differences only for small values of the overlap function. Navas-Guzman et al. (2011) and Rogelj et al. (2014) retrieve the overlap function of the total signal at 532 nm (sum of parallel and perpendicular channels) by means of the method presented by Wandinger et al. (2000). This study shows that the full-overlap height of MULHACEN is around 0.72 km agl. Assuming that the artefacts due to thermal fluctuations are negligible for overlap-function values above 70%,*

*depolarization profiles can be exploited in terms of MLH detection above ~0.25 km agl. Further details about the technical specifications of MULHACEN are provided by Guerrero-Rascado et al. (2008, 2009).'*

About the polarization properties, it has been included a reference (Bravo-Aranda, et al., 2016) where the systematic errors of the volume linear depolarization ratio determined with MULHACEN and other lidars in the EARLIENT network are explained in detail. Among others, the most important lidar parameter that can affect the depolarization retrieval is the diattenuation of the receiving optics which is well characterized in MULHACEN. However, these parameters affects to the depolarization calibration which is not required by POLARIS.

Figure: Rogelj et al. 2014: Experimental determination of UV- and VIS- lidar overlap function. Opt. Pura Apl. 47 (3) 169-175.

- Title: I think the title is not representative for the paper as it not only deals with POLARIS. Probably, the model comparison approaches and MWR should be accounted for in the title.

The title has been changed by '*A new methodology for PBL height estimations based on lidar depolarisation measurements (POLARIS): analysis and comparison against microwave radiometer and WRF model based results'.*

-Section 5 should be shortened: Please only show up with new information an restrict to WRF. Almost no new information with respect to the other already published results of R. Banks are given. Probably use WRF to check consistency of retrievals (begin of convection etc.,)

According to the suggestion of the Referee, we have shortened Section 5. Unfortunately, references about PBL height detection using WRF are mainly centered in suitable situations (no clouds, stable, aerosol-free free troposphere) and thus, the comparison with other studies become difficult. The beginning and end of the convection is discussed in the manuscript.

Specific comments (no spelling mistakes):
Page 5, line 11. Unit missing after 0.05 and later in the manuscript. Or what do you mean with dilation parameter?

The unit of the dilation parameter (i.e., kilometer). It has been specified through the manuscript.

Page 5, line 21: What do you mean: can be applied to lidars not fully characterized? I think this is a very "dangerous" statement.

We agree with the referee that the sentence is not appropriate. We meant that the depolarization calibration is not required. The sentence has been changed by: '*Since POLARIS is based on vertical relative changes, the depolarization calibration is not required facilitating the procedure*'.

Page 6: line 12: how can you assure, that this increase is not due to instrumental effects? The ratio of two signals very close to the lidar might not cancel out all instrumental effects anymore, especially regarding depolarization. Do you have a case study where it is seen that the depolarization ratio is constant throughout the atmosphere? I
always see an increase towards ground below 1 km.

The discussion about the overlap effect has been included in a previous comment.

Page 6, line: I did not understand what you mean with lowest layer, please rephrase: "Then, we define two layers: from the full-overlap height up to the lowest candidate, and from the lowest layer up to the highest candidate."

According to the comment, the phrase has been changed by: '*from 120 m agl up to the lowest candidate, and the layer between the lowest and the highest candidate*'.

Page 6, line 26: I do not know what you mean with aerosol stratification in this context, can you explain better?

We assume that the Referee makes reference to the Page 7/line 26. We have changed these phrases for the sake of clarity: '*According to Angelini et al. (2009), occasional aerosol stratification may occur within the mixing layer. This type of stratification which are usually short in time are not really linked with the planetary boundary development leading a false detections of the PBL height. A 7 bin moving median filter is used to reject the possible attributions related to this type of aerosol stratification.*'

Page 7, line 4 to 6. Are these thresholds really without dimension?

They are dimensionless. It has been specified through the manuscript.

Page 8, line 11: An offset of 300 m is not too small, how reliable are the MWR measurements during night time? And how reliable is your lowest candidate, as it is very close to the lidar. I wonder if the depolarization measurements are reliable (see comment above), please discuss this. Especially as you state that Polaris improves the detection. I cannot follow this discussion, as I do not know the "truth".

These differences can be explained considering the different tracers of both methods, and the fact that the POLARIS and MWR are detecting the residual and stable layer top, respectively. The discussion of this paragraph has been changed and a typo in the offset has been corrected: '*The offset of 600 m observed between $z_{SL}^{MWR}$ and $z_{RL}^{POL}$ during the night is mostly due to the fact that $z_{RL}^{POL}$ corresponds to the residual layer and $z_{RL}^{MWR}$ marks the top of the nocturnal stable layer.*'

Page 8, line 22. Between 11:20 and 11:30 UTC on which day? Again it is hard to follow, because the symbols are so tiny and especially the Polaris an (star) is almost invisible.

We have rephrased some sentences and included include the date to help the reader to follow the arguments: '*For example, on 16 June 2013, $z_{ML}^{MWR}$ increases from 0.8 km to 2.02 km agl between 10:15 and 11:30 UTC whereas $z_{ML}^{POL}$ increases abruptly from 0.52 to 1.82 km agl between 11:20 and 11:30 UTC (i.e., almost one hour later)*'. Also, the figure has been improved.

Page 8: Discrepancies between MWR and Polaris on 16 June afternoon should be more intensively discussed. I see a systematic bias of almost 1 km.

We agree with the Referee and the discussion has been improved including the difference noted by the Referee among others: '*For example, on 16 June 2013, $z_{ML}^{MWR}$ increases from 0.8 km to 2.02 km agl between 10:15 and 11:30 UTC whereas $z_{ML}^{POL}$ increases abruptly from 0.52 to 1.82 km agl between 11:20 and 11:30 UTC (i.e., almost one hour later). This is because $z_{ML}^{MWR}$ growths due to the increase of the temperature at surface level during the morning whereas $z_{ML}^{POL}$ increases later, once the convection processes are strong enough to dissipate the boundary between the mixing and the residual layer. Another example of the influence of the tracer is the 1-km bias between $z_{ML}^{POL}$ and $z_{ML}^{MWR}$ between 18:00 and 21:00 UTC on 16 June 2013. During the late afternoon and early night, the temperature at surface level quickly decreases and the atmospheric stability suddenly changes from instable to stable. This pattern is registered by the $z_{ML}^{MWR}$ decreasing from 1.82 km to 0.055 km agl between 18:00 and 18:30 UTC. The increasing atmospheric stability during the late afternoon and early night stops the convection processes and then the mixing layer becomes the residual layer. This change from mixing to residual layer is tracked by the temporal evolution of $z_{RL}^{POL}$ decreasing from 1.92 km to 0.52 km agl between 18:00 and 24:00 UTC. Therefore, there are differences between $z_{PBL}^{POL}$ and $z_{PBL}^{MWR}$ explained in terms of the tracer used for each method that are not related to a wrong attribution of POLARIS.*'

Page 9, line 22ff: Does the MWR really watch the same atmospheric column or is scanning included? Then the discussion could be different. Furthermore, also the overlap issue and the near-range depolarization issues could play a role.

HATPRO uses a combination of the scanning mode (measurements at 5.4°, 10.2°, 19.2°, 30°, 42° y 90°) near the surface, below 2 km, and zenithal mode above, since Crewell and Lohnert (2007) showed that these elevation scanning measurements increase the accuracy of the retrieved temperature, specifically in the boundary layer. Thus, we might initially consider that the MWR and the lidar are not exploring the same atmospheric column. However, the channels used during the scanning mode are more opaque (frequencies larger than 54GHz) and thus, the received radiation comes from the nearest layers (i.e., nearest to the zenithal observation). Conversely, the more transparent channels, providing information from the far field, are used during the zenithal observations. Taking into account this information, we do not think that the scanning mode is relevant for the analyses.

Page 11 line 31: These are certainly no long-term measurements, use better continuous measurements

Done.

Page 12, line 7ff: POLARIS allows better model validation compared to what?

We have rephrased the sentence: '*Since POLARIS allows detecting reliable PBL heights under Saharan dust outbreaks, it might be used for the improvement of the WRF parametrization.*'

Fig. 5 please move legend to case B and enlarge - it is not readable even at 150% zoom. Sometimes the blue star changes the color? If the star is overlaid with a dot please make it visible.

Done. Figure 5 has been improved.

Fig. 6 and 7. See above: Poor quality, work on improvement, otherwise discussion cannot be followed. Explanation for WRF missing in caption.

Done. Figure 6 and 7 has been improved.

Fig. 8: pink dot is somewhere else in legend

Done. Figure 8 has been improved.

Answers to Anonymous Referee #2:

The present version of Bravo-Aranda's paper, however, suffers from several short-comings. First, the small set of measurements is problematic. As a consequence, it cannot be demonstrated in a convincing way that POLARIS is useful when long time series shall be evaluated. Even for the short period of only 3 days (according to doi:10.5194/acp-16-455-2016, the SOP I of ChArMEx 2013 took place from 11. June to 5. July: what about this data-set?) a lot of situations were found when the different methodologies disagree. It is also not clear whether the basic assumption (if I understand it correctly, unfortunately it is not sufficiently explained) "a depolarizing aerosol layer is a transported desert dust layer and does not belong to the mixing layer" can be applied to other sites than Granada (or other Mediterranean countries). So this paper (provided the text has been undergone a substantial revision in grammar and spelling, and has been improved in terms of scientific clarity) can only be considered as a first contribution to a discussion of the benefit of adding depolarization-information to a MLH-retrieval.

Despite the ChArMEx campaigns took place from 11 June to 5 July 2013, the continuous lidar measurements were scheduled from 9 to 11 July 2012 (~72 hours) and from 16 to 17 June 2013 (~36 hours). We use these continuous lidar measurements under Saharan dust outbreaks to analyze the temporal evolution of the PBL. Also, the periods are large enough to demonstrate how POLARIS improves the detection with respect to the method which uses only the RCS and to evidence that the depolarization measurements is useful for the detection of the PBL height.

Despite POLARIS is validated using dust layers coupled to the PBL, a priori, it can be used for any layer coupled to the PBL if the aerosol particle-shape is different enough to be detected by the depolarization profile. For example, a dust layer coupled to the PBL which is a frequent scenario in all the Mediterranean countries as well as in those regions affected by dust outbreaks. POLARIS represents a real improvement compared to previous methodologies used for the automatic detection of the PBL height based on lidar data using only the RCS, which tended to erroneously estimate the PBL top under complex situations. As observed in the manuscript, much lower differences are observed between $z_{PBL}^{POL}$ and the reference ($z_{PBL}^{MWR}$) than between $C_{RCS}$ and $z_{PBL}^{MWR}$. Nonetheless, discrepancies still exist between the PBL height determined with the MWR and POLARIS, but they can be easily explained taking into account the use of different tracers (i.e. temperature and aerosol) and the uncertainties associated to both methodologies. The text in the manuscript has been modified accordingly.

I will not list the countless typos, word repetitions, cases of wrong grammar, misuse of capital letters, misspelled units, or undefined symbols. Even one of the affiliations is not correct and the link to the ChArMEx-Website does not work! Only a few specific mostly sci        ence-related issues are listed below. In summary I think the authors should take all comments seriously. If not all issues are fully resolved I will not recommend the publication of a revised version.

Specific comments:

• page 2, lines 14ff: There is an extensive discussion on different regimes as the residual layer, the mixing layer, the convective boundary layer and more. In the rest of the paper primarily the "PBL" is mentioned and discussed (see 3/5). A strict terminology is required throughout the paper. When there is a co-existence of the residual layer and the convective layer (e.g. after sunrise) PBL might be confusing.

The advice of the Referee has been taking into account and thus, the discussion of the results has been updated according to the PBL type (e.g., mixing layer, residual layer and stable layer) through the manuscript.

• 2/23: "Sunrise and Sunset are characterized by the complexity of the PBL.": This sentence is really strange!

We agree with the referee that the sentence is not appropriated. The sentence has been changed by: '*The PBL structure is especially complex during the sunrise and sunset when the mixing and residual layers may coexist*'.

• 3/18: "... are feasible and reliable ...": What is meant with "reliable"? In the paper many example are shown when this is not the case.

The word 'reliable' has been removed and the text has been rewritten: '*Since the experimental detection of $z_{PBL}$ is spatially and temporally limited due to instrumental coverage, the use of Numerical Weather Prediction (NWP) models for the estimation of $z_{PBL}$ is a feasible alternative*'.

• 3/21: "... include stringent conditions ...": What is this?

The text has been modified and thus, the word 'stringent' has been removed: '*Since the experimental detection of $z_{PBL}$ is spatially and temporally limited due to instrumental coverage, the use of Numerical Weather Prediction (NWP) models for the estimation of $z_{PBL}$ is a feasible alternative. In this regard, several validation studies of these model estimations have been conducted based on lidar and surface and upper air measurements (Dandou et al., 2009; Helmis et al, 2012), some of them in areas close to the study region (Borge et al., 2008; Banks et al., 2015). Results showed that NWP estimations of the $z_{PBL}$ ($z_{PBL}^{WRF}$) are feasible, but with a tendency to the underestimation of the $z_{PBL}$ in most synoptic conditions. In this study, $z_{PBL}^{WRF}$ is tested against the $z_{PBL}$ derived from POLARIS and MWR measurements under Saharan dust events*'.

• 4/8: There are a lot of words on the overlap, but the most relevant number, i.e. the minimum range that can be exploited in terms of MLH, is missing. Why is the 90% overlap-range given as a interval? Is it temperature dependent? If it depends on the channel (i.e., wavelength) but only one wavelength is used in this study, it is not adequate to give a range. It is strange that the polarization channels which are the most relevant in view of the novelty of POLARIS are not mentioned here – whereas the irrelevant Raman- channels and water vapor channels are mentioned.

We agree with the Referee. We have included more information about the overlap of the polarizing channels and discussed the possible influences on the measurements as follows: '*The optical path of the parallel and perpendicular channels at 532 nm are designed to be identical up to the PBC where the 532 nm signal is split into parallel and perpendicular before reaching the PMT. This setup allows us to assume almost the same overlap for both polarizing components. Thus, the depolarization profile is practically not influenced by the incomplete overlap since it is cancelled out by the ratio of the perpendicular and parallel channels. Only the thermal dilation and contraction of the lidar optics after the PBC might independently change the overlap function of each channel. Since MULHACEN is deployed inside an air-conditioned building, the temperature fluctuation is small and thus, the overlap difference between the channels might be low. Therefore, we assume significant differences only for small values of the overlap function. Rogelj et al. (2014) retrieve the overlap function of the total signal at 532 nm (sum of parallel and perpendicular channels) by means of the method presented by Wandinger et al. (2000). This study shows that the full-overlap height of MULHACEN is around 0.72 km agl. Assuming that the artefacts due to thermal fluctuations are negligible for overlap-function values above 70%, depolarization profiles can be exploited in terms of MLH detection above ~0.25 km agl. Further*

*details about the technical specifications of MULHACEN are provided by Guerrero-Rascado et al. (2008, 2009).'*

• 4/28: The vertical resolution given here does not agree with the statement in line 21.

We have removed the vertical resolution on that sentence and rephrased that paragraph as follows: *'The MWR temperature profile is used to locate the $z_{PBL}$ ($z_{PBL}^{MWR}$) by two algorithms. Under convective conditions, fuelled by solar irradiance absorption at the surface and the associated heating, the parcel method is used to determine the mixing layer height $z_{ML}^{MWR}$ (Holzworth, 1964). Granados-Muñoz et al. (2012) already validated this methodology obtaining good agreement with radiosonde measurements. Since the parcel method is strongly sensitive to the surface temperature (Collaud-Coen et al., 2014), surface temperature data provided by the MWR are replaced by more accurate temperature data from a collocated meteorological station, in order to minimize the uncertainties in $z_{ML}^{MWR}$ estimation. Conversely, under stable situations, the stable layer height $z_{SL}^{MWR}$ is obtained from the first point where the gradient of potential temperature ($\vartheta$) equals zero. Collaud-Coen et al. (2014) determine the uncertainties of the PBL height for both methods by varying the surface temperature by ±0.5°. The uncertainties are on the order of ±50 to ±150 m for the PBL maximum height reached in the early afternoon, although uncertainties up to ±500 m can be found just before sunset. Further details about both methods are given by Collaud-Coen et al. (2014).'*

• 5/24: "both … are normalized respectively to the maximum value of RCS and δ in the first kilometer above the surface ". In case of δ no normalization can be seen in any of the corresponding figures. Please clarify.

The correct normalized δ profile has been included in all the axes of Figure 5.

• section 3: It is not common to use the character "C" for a height.

We agree that the character is not commonly used as height but we need to distinguish between the candidates and the final PBL height. Thus, we decide to use 'C' for the candidates where the subscript makes reference to the type of candidate as listed in the point 2) of the Section 3.2:

  i.   $C_{RCS}$: *the height of the $W_{RCS}$ maximum closest to the surface exceeding a certain threshold $\eta_{RCS}$. This threshold is iteratively decreased, starting in 0.05, until $C_{RCS}$ is found. This is procedure established by Granados-Muñoz et al. (2012). A dilation value ($a_{RCS}$) of 0.03 km is used according to Granados-Muñoz et al. (2012).*
  ii.  $C_{min}$: *the height of the $W_\delta$ minimum closest to the surface exceeding the threshold $\eta_{min}$. $C_{min}$ indicates the height of the strongest increase of δ.*
  iii. $C_{max}$: *the height of the $W_\delta$ maximum closest to the surface exceeding the threshold $\eta_{max}$. $C_{max}$ indicates the height of the strongest decrease of δ.*

and 'z' is used for the final PBL where the superscript makes reference to the method (i.e., 'POL', 'MWR', 'WRF'). We think that the sentences in Section 2 adequately highlights that the symbol 'C' makes reference to the height of the candidates.

- 6/4: Fig. 6 is discussed prior to Figs. 4 and 5. Fig. 3 is missing!

We have changed the numeration of the figures.
Page 4/L19: Fig. 1
Page 4/L20: Fig. 2
Page 5/L12: Fig. 3
Page 5/L15: Fig. 4
Page 6/L18: Fig. 5
Page 6/L26: Fig. 6
Page 7/36: Fig. 7

- 6/5: What is the reason for selecting RCS at 532 nm? Furthermore, "height above mean sea level" should be transformed to "height above ground" (throughout the paper).

We decided to plot the RCS and the $\delta$ at the unique wavelength with depolarization capability (532 nm).

- 6/8: "We do not expect the ...": Does this mean that an automated POLARIS retrieval is not possible?

The sentence is confusing and thus, it was removed. Additionally, we have move the point 3) from the methodology to the end of the Section 3.2 where we illustrate how the distribution in height of the candidates is related to a specific atmospheric situation as follows: '*To illustrate how the distribution in height of the candidates is related to a specific atmospheric situation, we analyse a particular case at 21:30 UTC on 16 June 2013 (Fig. 5) corresponding to an example of the c.1 scenario. As can be seen, $C_{RCS}$ and $C_{max}$ are located at 4.46 and 4.41 km agl whereas $C_{min}$ is located at 0.7 km agl. Since the different between $C_{RCS}$ and $C_{max}$ is lower than 0.15 km, we assume that both candidates points to the same edge of the layer and thus, this situation corresponds to $C_{RCS} = C_{max} > C_{min}$. The mean and variance of $\delta$ in the layer below $C_{min}$ ands the layer between $C_{min}$ and $C_{max}$ are 0.65 and 7·10-4 and 0.99 and 91·10-4, respectively. Since the $\delta$ mean difference is larger than $\delta_t$ and the variances differ more than 30%, we determine that there are two different layers: the PBL (low $\delta$) and the coupled layer (high $\delta$) where $C_{RCS} = C_{max}$ indicates the coupled layer top and $C_{min}$ indicates the limit between the residual and the coupled layer, being chosen as $z_{PBL}$. In this particular case, POLARIS improves the $z_{PBL}$ detection from 4.46 agl to 0.7 km agl*'.

- 6/15ff: The following discussion is confusing. A few examples: Under "b.1" it is stated that CRCS =Cmax (by the way another typo: should be C RCS) whereas in the next line of text the authors describe that C CRS =Cmax! It is not clear, what the "lowest layer" is (line 22). It is doubtful that

at the top of a lofted layer RCS increases (7/2), the opposite should be the case. It is not clear why there is an increase of δ "before C max" (7/7). It should be clearly outlined what should be understood by "coupled", it seems that it is used in different ways.

The whole Section 3.2 has been rewritten according to the comments of the Referee #2.

It is difficult to understand a situation when C min > Cmax, whereas the opposite can be identified as e.g. a lofted dust layer.

$C_{min} > C_{max}$ means that $\delta$ profile has an abrupt decrease ($C_{min}$) and then, an abrupt increase ($C_{max}$). This pattern fits with the presence of a lofted layer above the PBL (as considered in the scenario c.2.2). For example, at 06:00 UTC on 11/07/2012 (Fig. 7).

• In Fig. 5 the differences of the profiles in cases D/E or F/G are hardly visible. Nevertheless the retrieval results in quite different zPBL. This seems to be a weakness of the method and should be discussed in detail. Moreover, case I
seems to be critical. The "inhomogeneities" in the shape of the δ-profiles are not much pronounced so it seems questionable if depolarization should be exploited at all, especially when considering measurement errors (error bars are missing in all figures!). The labels of the axes and the legend are hardly readable.

Figures have been improved to facilitate the analysis. The inhomogeneities in the shape of the δ-profiles are strong enough to allow the detection of at least one of the candidates $C_{min}$ and $C_{max}$. In any case, the absence of both candidates should not be considered as a weakness of the method but a possible scenario in which the depolarization changes are not stronger enough to find an edge and thus, $C_{RCS}$ is chosen as $z_{PBL}$. The methodology has been changed to include this possibility. However, this change does not affect to the results presented in the manuscript because this situation did not occur. The error bars have been omitted since we do not performed a direct comparison between the profiles. In fact, these figures just illustrate the different candidate height distributions of each scenario.

• Section 3.3: A discussion of how the different thresholds are found is missing. There are only statements on specific numbers. The rest of the text does not really fit to the title of the section; it is rather a discussion of the differences of the old and new method.

More comments about the optimization process has been included to fit the discussion to title of the section.

• 8/16: "CRCS indicates layeringpoints to a weak edge within the PBL. " Another example of a "weird" sentence.

The Section 3.2 has been rewritten according to the comments of the Referee #2.

• 8/21: Why do the authors switch to "m" instead of "km" as in the rest of the text?

We have decided to switch from m to km in the whole manuscript.

• Section 4: Validation is performed by means of the MWR-retrieval. This implies that the latter is assumed to be the truth (see analysis in doi:10.5194/amt-7-3685-2014). As a consequence the MWR-retrieval and its accuracy has to be explained in more detail.

In this study, the optimization and validation of a new methodology to determine the PBL height has to be performed against a PBL height derived from independent measurements. Thus, we use the PBL height derived from MWR temperature profiles. We have modified the manuscript to highlight that the reference PBL includes also uncertainties and weak points in the methodology. Additionally, more information about its accuracy has been included.

In Section 4 the authors demonstrate that there are a lot of differences. Thus, the reader might conclude that the POLARIS-retrieval does not work reliably (in my view the grey and black stars never coincide in Fig. 7). In Fig. 7 it is not explicitly explained which parameter is shown in the upper/lowerpanel.

Figure 7 has been corrected.

• 9/24: There are no red triangles in Fig. 9!

Done. Figure 8 has been improved.

• Section 5: Obviously there are very few cases when POLARIS-retrievals agree with the WRF-simulations. What is the conclusion with respect to the usefulness of POLARIS or the accuracy of WRF?

As reported in the manuscript there has been previous studies evaluating the performance of the WRF model estimating the PBL height. Nevertheless, few of them, if any, have evaluated this performance under the complex conditions here analyzed. As commented in the manuscript, the main differences are found during daytime under the Saharan dust outbreaks, were the WRF model clearly underestimates the PBL height.  Although other reason can explain this underestimation (as for instance insufficient number of model layers), the more plausible one is the inability of the here used WRF PBL parameterization to account properly for this particular kind of events. This is the main conclusion regarding the POLARIS and the WRF model. Thus, in future work, other PBL parameterizations may be evaluated. In addition, the here used dataset and the POLARIS method may be used to improve the WRF PBL parameterizations.